# Extracellular Nicotinamide Phosphoribosyltransferase Is a Therapeutic Target in Experimental Necrotizing Enterocolitis

**DOI:** 10.3390/biomedicines12050970

**Published:** 2024-04-28

**Authors:** Melissa D. Halpern, Akash Gupta, Nahla Zaghloul, Senthilkumar Thulasingam, Christine M. Calton, Sara M. Camp, Joe G. N. Garcia, Mohamed Ahmed

**Affiliations:** 1Division of Neonatology, Department of Pediatrics, College of Medicine, University of Arizona, Tucson, AZ 85724, USA; 2Center for Inflammation Science and Systems Medicine, University of Florida Scripps Research Institute, Jupiter, FL 33458, USAjgn.garcia@ufl.edu (J.G.N.G.)

**Keywords:** necrotizing enterocolitis, NEC, extracellular nicotinamide phosphoribosyltransferase, eNAMPT, NAMPT, DAMP, ALT-100, TLR4, TGFβ

## Abstract

Necrotizing enterocolitis (NEC) is the most common gastrointestinal emergency of prematurity. Postulated mechanisms leading to inflammatory necrosis of the ileum and colon include activation of the pathogen recognition receptor Toll-like receptor 4 (TLR4) and decreased levels of transforming growth factor beta (TGFβ). Extracellular nicotinamide phosphoribosyltransferase (eNAMPT), a novel damage-associated molecular pattern (DAMP), is a TLR4 ligand and plays a role in a number of inflammatory disease processes. To test the hypothesis that eNAMPT is involved in NEC, an eNAMPT-neutralizing monoclonal antibody, ALT-100, was used in a well-established animal model of NEC. Preterm Sprague–Dawley pups delivered prematurely from timed-pregnant dams were exposed to hypoxia/hypothermia and randomized to control—foster mother dam-fed rats, injected IP with saline (vehicle) 48 h after delivery; control + mAB—foster dam-fed rats, injected IP with 10 µg of ALT-100 at 48 h post-delivery; NEC—orally gavaged, formula-fed rats injected with saline; and NEC + mAb—formula-fed rats, injected IP with 10 µg of ALT-100 at 48 h. The distal ileum was processed 96 h after C-section delivery for histological, biochemical, molecular, and RNA sequencing studies. Saline-treated NEC pups exhibited markedly increased fecal blood and histologic ileal damage compared to controls (*q* < 0.0001), and findings significantly reduced in ALT-100 mAb-treated NEC pups (*q* < 0.01). Real-time PCR in ileal tissues revealed increased NAMPT in NEC pups compared to pups that received the ALT-100 mAb (*p* < 0.01). Elevated serum levels of tumor necrosis factor alpha (TNFα), interleukin 6 (IL-6), interleukin-8 (IL-8), and NAMPT were observed in NEC pups compared to NEC + mAb pups (*p* < 0.01). Finally, RNA-Seq confirmed dysregulated TGFβ and TLR4 signaling pathways in NEC pups that were attenuated by ALT-100 mAb treatment. These data strongly support the involvement of eNAMPT in NEC pathobiology and eNAMPT neutralization as a strategy to address the unmet need for NEC therapeutics.

## 1. Introduction

Despite advances in neonatal practice, necrotizing enterocolitis (NEC) remains the most common gastrointestinal emergency in premature infants, with an incidence of 6–10% in preemies weighing less than 1500g at birth [1]. NEC is characterized by hemorrhagic inflammation of the ileum and colon [2], with mortality of 20–30% [3,4]. Clinical presentation ranges from abdominal distension to intestinal gangrene and necrosis [2], with surgical intervention often resulting in short bowel syndrome and also being a significant predictor of neurodevelopmental morbidity [5,6]. The economic costs of NEC are substantial; surgical NEC is one of the costliest morbidities of premature infants [7]. While the major risk factors for NEC—e.g., prematurity, formula feeding, and dysbiosis of the intestinal microbiome—are known [8], treatment strategies beyond laparotomy are predominantly supportive [9,10,11].

While the pathogenesis of NEC remains poorly understood, disruption of the intestinal barrier during the early stages of NEC development is thought to be crucial [12,13]. Gastrointestinal barrier failure can lead to translocation of Gram-negative bacteria and bacterial wall products, such as lipopolysaccharide (LPS), into circulation to initiate the proinflammatory cascade observed in NEC [14,15], including tumor necrosis factor alpha (TNFα) [16,17], interleukin-6 (IL-6) [18,19], and interleukin-8 (IL-8) [15,20,21]. LPS binding and activation of the pathogen recognition receptor Toll-like receptor 4 (TLR4) are essential to innate immunity regulation and result in marked NF-κB-driven inflammatory cytokine production [22,23]. TLR4 is well recognized as a major participant in NEC [24,25,26,27,28,29]. Infants with NEC have increased expression of intestinal TLR4 compared to healthy preterm infants [26]. Additionally, experiments in animal models of NEC have shown that inactivation of TLR4 signaling protects against disease development [24,25]. Intestinal transforming growth factor beta (TGFβ) is also implicated as a casual factor in NEC. TGFβ is an important family of growth factors that regulate development, wound healing, and immune function [30]. The expression of TGFβ is downregulated in patients with NEC [31,32,33], and treatment with exogenous TGFβ protects against disease in an experimental model of NEC [32]. Interestingly, LPS-activated TLR4 can inactivate TGFβ signaling via crosstalk between TLR4 signaling and the canonical Wnt/β-catenin pathway [34,35,36].

Human breast milk contains components that modulate a variety of immune pathways, including both TLR4 and TGFβ signaling [37,38]. Breast milk contains epidermal growth factor, which inhibits activation of intestinal TLR4. This inactivation results in reduced apoptosis of the intestinal epithelia and increased enterocyte proliferation [39]. Breast milk also contains large quantities of TGFβ, which has been shown to decrease inflammation in the neonatal intestine via Erk and SMAD6 [40]. These findings help explain the strong protective effect that breast milk has in NEC [41,42,43]. However, there are currently no treatments that can mitigate TLR4 signaling and promote TGFβ expression once NEC has developed.

In addition to LPS, a number of proteins bind TLR4 with varying levels of cascade activation, including damage-associated molecular pattern (DAMP) proteins that serve as host defense sentinels [44,45,46]. DAMPs contribute to the severity of multiple inflammatory and fibrotic disorders via ligation of pathogen-recognition receptors (PRRs), such as the Toll-like receptor family, and subsequent activation of evolutionary-conserved inflammatory pathways. We previously used genomic strategies and identified extracellular nicotinamide phosphoribosyltransferase (eNAMPT) as a novel DAMP and TLR4 ligand that potently initiates NF-κB transcription-dependent inflammatory responses and cytokine release [47]. We have shown that the eNAMPT/TLR4 inflammatory cascade is a druggable pathway in preclinical models of pulmonary hypertension [48,49,50], acute respiratory distress syndrome [51], radiation-induced lung disease [52,53], lupus vasculitis [54], and Crohn’s disease [55,56].

Because eNAMPT has been shown to activate TLR4—which is crucial to NEC development—we hypothesized that treatment with an eNAMPT-neutralizing mAb would protect against NEC. The objective of this study was to test whether the humanized eNAMPT-neutralizing mAb ALT-100 (43)—currently in phase 2A acute respiratory distress syndrome (ARDS) clinical trials (ClinicalTrials.gov, NCT05938036)—can protect against NEC development in a well-established animal model of this disease [16,57,58,59]. The data presented herein show, for the first time, the involvement of eNAMPT in NEC pathobiology and point to eNAMPT neutralization as a potential novel strategy to address the unmet need for effective NEC therapeutics.

## 2. Methods

Antibodies. Antibodies immunoreactive against b-actin were purchased from Invitrogen (Carlsbad, CA, USA). Goat, rabbit, and mouse secondary antibodies were purchased from Life Technologies (Waltham, MA, USA). Human PBEF/visfatin biotinylated antibody (Cat #BAF4335) was purchased from R&D Systems (Minneapolis, MN, USA), NAMPT monoclonal antibody from Thermo Fisher (Waltham, MA, USA, Cat # 66385-1-IG), ultra streptavidin-HRP (Thermo Fisher, Cat #N504), and actin-HRP (Cat #A3854-200UL) from Sigma-Aldrich (St. Louis, MO, USA). Peroxidase Affini Pure goat anti-mouse IgG (H + L) (Cat #102646-170) and anti-IgG (H + L) goat polyclonal antibody (horseradish peroxidase) (Cat #102645-188) and IgG for use as controls were obtained from Jackson ImmuneResearch (West Grove, PA, USA). The eNAMPT-neutralizing humanized mAb ALT-100 [54] was provided by Aqualung Therapeutics Corporation (Juno Beach, FL, USA).

NEC protocol. All protocols were approved by the Animal Care and Use Committee of the University of Arizona (D16-00159 A-3248-01). All animals were monitored for signs of abdominal distension, respiratory difficulty, or weight loss and euthanized prior to the studies’ end if any of these parameters were excessive.

Experimental study and applied design. To develop NEC, a well-characterized rat model was used [16,57,58,59]. The formula/hypoxia/hypothermia model of NEC was chosen because it most closely mimics the risk factors for NEC: prematurity, formula feeding, and stress (modeling apnea of prematurity [60,61] a common condition of premature infants, and the inability of preemies to adequately regulate temperature [62,63]. Specifically, for each of the two separate studies, pups from 3–5 timed-pregnant Sprague–Dawley rats (Charles River, Hollister, CA, USA) were delivered via C-section one day prior to scheduled birth. Pups were divided into four groups, each containing the same number of pups from each litter and with similar mean birth weights per group.

Control + vehicle (Ctrl), *n* = 10: fed by a foster dam and injected with 50 μL of saline 48 h post-deliveryControl + ALT-100 (Ctrl + mAb), *n* = 10: fed by a foster dam and injected with 10 µg of ALT-100 diluted in 50 μL of saline 48 h post-deliveryNEC + vehicle (NEC), *n* = 29: exclusively hand-fed via oral gavage with 200 μL of formula (2:1 Similac Advance powder, Abbott Nutrition, Columbus, OH, USA) and Esbilac Puppy Milk Replacer, Pet-Ag, Hampshire, IL, USA) 5 times per day and injected with 50 µL of saline 48 h post-deliveryNEC + ALT-100 (NEC + mAb), *n* = 28: exclusively hand-fed via oral gavage with 200 μL of formula 5 times per day and injected with 10 µg of ALT-100 in 50 µL of saline 48 h post-delivery

Pups from all four groups were exposed to hypoxia (N_2_ gas for 60 s), followed by hypothermia (4 °C for 10 min) twice daily [16,57,58,59] (Figure 1).

Data source. All studied pups were euthanized at 96 h post-delivery, and trunk blood was collected for serum analyses. A section of the distal ileum was removed and fixed in paraformaldehyde and then paraffin-embedded for H & E staining, and a separate, adjacent section of the ileum was flash-frozen for RNA sequencing and Western blotting.

Procedure: histological scoring of NEC. Pathologic changes in the intestinal architecture were evaluated using our previously published NEC scoring system. Histological changes were scored by a blinded evaluator and graded as follows: 0 (normal)—no damage; 1 (mild)—slight submucosal and/or lamina propria separation; 2 (moderate)—moderate separation of the submucosa and/or lamina propria, and/or edema in submucosal and muscular layers; 3 (severe)—severe separation of the submucosa and/or lamina propria, and/or severe edema in submucosal and muscular layers and regional villous sloughing; and 4 (necrosis)—loss of villi and necrosis. Intermediate scores of 0.5, 1.5, 2.5, and 3.5 were also used to more accurately assess ileal damage. Animals with scores below 2.0 were considered to be NEC-free [64,65,66].

Occult blood in feces. Feces were obtained at 72 h post-delivery during urogenital stimulation prior to feeding. Occult blood was determined via the guaiac test, as previously described, and scored on a scale of 0–4 [58].

Serum cytokine levels. Cytokine levels in rat serum were analyzed for levels of eNAMPT, IL-6, serum chemokine C-X-C motif ligand 1 (CXCL1), and TNFα using the U-Plex MSD ELISA platform (Meso Scale Diagnostics, Rockville, MD, USA), as previously described [49,52,53,54,67].

TGFβ levels in ileal tissues. Ileal tissues from control + mAb, NEC, and NEC + mAb groups were homogenized in T-PER tissue lysis buffer (Thermo Fisher Scientific, Carlsbad, CA, USA), and the samples were quantified to determine the protein concentration using the BCA protein estimation kit (Thermo Fisher Scientific). Legend Max Total TGFβ1 ELISA kit assay (Biolegend Inc., San Diego, CA, USA) was performed to determine TGFβ1 levels normalized to the protein concentration per µg. ELISA plate data were calculated with GainData software (Arigo Biolaboratories, Zhubei City, Taiwan) using a 4-parameter logistics curve-fitting algorithm. The mean absorbance for each set of triplicate standards was obtained.

NAMPT Western blotting. The snap-frozen ileum was homogenized in RIPA buffer (50 mmol/L Tris-HCl, pH 7.4, 150 mmol/L NaCl, 0.5% sodium deoxycholate, 0.1% SDS, 1% NP-40, 5 mmol/L EDTA) supplemented with a complete protease/phosphatase inhibitor cocktail (Cell Signaling, Danvers, MA, USA, Cat #5872S) using a tissue grinder with glass pestles (VWR, Radnor, PA, USA, Cat #26307-606). After centrifugation (15,000× *g* for 20 min at 4 °C), the protein concentration of homogenates was determined by Bio-Rad DC protein assay (Hercules, CA, USA, Cat #5000112). Following a 5 min incubation at 90 °C in loading buffer, aliquots containing equal amounts of protein (25–30 μg) were subjected to sodium dodecyl sulfate–polyacrylamide gel electrophoresis (SDS-PAGE). Subsequently, proteins were transferred to PVDF membranes and probed with specific primary and then with secondary antibodies. Proteins were visualized using an ECL system (Pierce West Pico, Cat #34580) and an ChemiDoc MP imaging system (Bio-Rad, Hercules, CA, USA). Densitometric analysis was performed using Bio-Rad Image Lab 6.01.

RNA sequencing of murine ileal tissue samples. Ileal RNA was extracted, and RNA was QC-assessed by the RIN value, 28S/18S and fragment length distribution (Aligient 2100 Bio analyzer, Agilent RNA 6000 Nano Kit, Santa Clara, CA, USA). Following library construction, RNA was sequenced using the Illumina Hiseq (NovaSeq) PE150 platform (San Diego, CA, USA), generating an average of 6 Gb raw data per sample. RNAseq data bioinformatic analyses included data quality control and calculation of Pearson correlations of all genes expressed to reflect the correlation of gene expression between samples. The Hierarchical Indexing for Spliced Alignments of Transcripts Bowtie2 (HISAT2) program was used to align and clean reads to the reference genome and to the reference genes [68,69,70]. The abundance and distribution of transcripts were assessed, obtaining the expected number of fragments per kilobase per million base pairs (FPKM) [71]. Correlation analysis to assess the variation between samples was performed by Pearson correlation. Deseq2 algorithms were used to detect DEGs with Bioconductor software packages, version 3.18 [72]. To control for multiple testing errors, the adjusted *p*-value false discovery rate was used [73]. Enrichment analysis for Gene Ontology (GO) classification was performed, focusing on biological process and pathway classification with KEGG and Reactome sources [74].

Statistical analysis. Statistical tests were performed using GraphPad Prism version 7.00 for Windows (GraphPad Software, La Jolla, California, USA). Outliers were determined by the ROUT method (Q = 1%). Differences in NEC pathology and occult blood were analyzed with the Kruskal–Wallis test for nonparametric values, followed by the Benjamini–Kreiger–Yekutieli false discovery rate method for multiple comparisons. Differences between groups were considered statistically significant when *q* values were less than 0.05 (*q* < 0.05). All other data were analyzed using standard one-way ANOVA, and groups were compared using the Newman–Keuls test. Differences between groups were considered statistically significant when *p*-values were less than 0.05. Principal component analysis (PCA) was derived based on the maximum variance in transcriptomic data and generated by the Partek algorithm, version pE150.

## 3. Results

### 3.1. Elevated NAMPT Expression in NEC Pup Blood and Ileal Tissues Is Attenuated by an eNAMPT-Neutralizing mAb

To determine whether NAMPT expression is altered during NEC, premature rat pups were delivered via cesarean section and subjected to a well-established protocol for inducing NEC in rodents. Experimental pups (NEC and NEC + mAb) were exclusively hand-fed formula, whereas control pups (Ctrl and Ctrl + mAb) were fostered by a lactating dam. All pups underwent hypoxia/hypothermia stress twice daily (Figure 1). Previous studies have shown that control pups do not develop ileal damage, whereas formula-fed pups exposed to hypoxia and hypothermia develop significant ileal pathology consistent with human NEC [16,57,58,59]. As proof of concept, we examined changes in serum eNAMPT levels in dam-fed, stressed pups and formula-fed littermates exposed to hypoxia and hypothermia. Figure 2A shows that serum eNAMPT levels were significantly elevated in experimental NEC (*p* < 0.05), a finding corroborated by significant elevations in both *NAMPT* mRNA levels (*p* < 0.05) (Figure 2B) and NAMPT protein immunoreactivity in ileal tissues (*p* < 0.05) (Figure 2C). NAMPT expression levels in blood and ileal tissues were significantly reduced in the NEC + mAb group (*p* < 0.01) (Figure 2B,C).

### 3.2. An eNAMPT-Neutralizing mAb Attenuates Ileal Tissue Pathology in Experimental NEC

Using our previously reported histologic scoring system [16,57,58,59], ileal damage was assessed [16,58,59] via histologic ileal damage in the four experimental groups. Both Ctrl and Ctrl + mAb groups had no histological damage consistent with NEC, indicating the eNAMPT-neutralizing mAb alone does not affect the ileal architecture. In contrast, the NEC group displayed statistically significant increased ileal damage compared to either control group (*q* < 0.0001 vs. Ctrl and Ctrl + mAb). Ileal damage scores in the NEC + eNAMPT mAb group were significantly reduced compared to NEC alone (*q* < 0.01) (Figure 3A). A similar pattern was observed in fecal occult blood scores, with the NEC + eNAMPT mAb group exhibiting significantly lower scores than the NEC group (*q* < 0.01) (Figure 3B). These results show that injection with the anti-eNAMPT mAb reduces the severity of experimental NEC. The representative histology for each group is shown in panels Figure 3C–F.

### 3.3. eNAMPT-Neutralizing mAb Decreases NEC-Induced Increases in Serum Proinflammatory Mediators

Proinflammatory mediators play an important role in NEC pathogenesis, including tumor necrosis factor alpha (TNFα) [16,75], IL-6 [33,76], and IL-8 [77,78]. Figure 4 depicts the elevated serum levels of (A) TNFα, (B) IL-6, and (C) IL-8 in NEC (*p* < 0.05 vs. Ctrl + mAb) and their reduction in rat pups that received ALT-100, the eNAMPT-neutralizing mAb (*p* < 0.01 vs. NEC).

### 3.4. RNA Sequencing Analysis of NEC Ileal Tissues and TLR4 and TGFβ Pathway Analyses

RNA sequencing of NEC ileal tissues from the four experimental NEC groups with principal component analysis (PCA) revealed differentially expressed genes (DEGs)/pathways associated with innate immune responses, cytotoxicity, inflammation, NK-mediated immunity, and autoimmunity between NEC and NEC + eNAMPT mAb groups (Appendix A). PCA of all genomic data showed clear separation by PC1 = 64.75%, PC2 = 19.59%, and PC3= 10.51% (Appendix A) and showed distinct sample variance at gene levels, as illustrated by Pearson’s sample similarity matrix (Appendix A). Differential Deseq2 algorithm analysis of normalized data between groups revealed DEGs associated with genes/pathways regulated with innate immune responses, cytotoxicity, inflammation, NK-mediated immunity, and autoimmunity.

The eNAMPT mAb blocks inflammation by inhibiting the activation of innate immune pathways via TLR4 [49,51,52,53,54]. Given that the inactivation of TLR4 has been shown to protect against NEC in experimental models of the disease [25,26,79], we hypothesized that treatment with the eNAMPT mAb would result in altered TLR signaling. The normalized gene count heat map for TLR signaling gene enrichment showed downregulation of genes that negatively affect TLR signaling, leading to its activation in the NEC group, while in the NEC + mAB group, the expression of these genes was reinstated, indicating its suppressive effect on the negative regulation of TLR signaling pathway activation (Figure 5A). The gene expression of *Ccl5* (Figure 5B), *Birc3* (Figure 5C), and *Oasl* (Figure 5D) was reduced in the NEC group (*p* = 1.2 × 10^−16^, 6.41 × 10^−4^, and 7.9 × 10^−6^, respectively) and increased in NEC pups treated with the eNAMPT mAb (*p* = 3.07 × 10^−5^, 9.03 × 10^−5^, and 7.24 × 10^−4^, respectively). *Cish* (Figure 5E) was also upregulated in the NEC + mAb group (*p* = 1.0 × 10^−4^) but without any difference between Ctrl + mAb and NEC groups. DEG-related pathways in the NEC + mAb group revealed upregulation of ileal inflammatory cytokine repressor genes *Atf3* (Figure 5F), *Fos* (Figure 5G), and *Jun* (Figure 5H) compared to the NEC group (*p* = 2.48 × 10^−48^, 2.77 × 10^−7^, and 5.03 × 10^−5^, respectively) and negative regulation of TLR and the non-canonical NF-κB pathway in the NEC group.

Principal component analysis of all genomic data in the Ctrl + mAb, NEC, and NEC + mAb groups showed excellent separation with dramatic differential gene expression between the NEC and NEC + mAb groups (Appendix A). The heat map for TGFβ receptor signaling showed changes in Atf3, Jun, and Fos (Figure 6A), which were also upregulated in the TLR4 analyses. Gene enrichment studies showed positive regulation of the TGFβ receptor signaling pathway in the NEC + mAb group compared to the NEC group, as depicted by the heat map in Figure 6A. Expression of the TGF*β* pathway gene *Foxo4* (Figure 6E) was reduced in the NEC group compared to the Ctrl + mAb group (*p* = 1.88 × 10^−4^). The expression of *Foxo4*, as well as *Mmp12* (Figure 6B), *Fosb* (Figure 6C), *Dynlrb2* (Figure 6D), *Klf6* (Figure 6F), and *Itgb4* (Figure 6G), was markedly increased in the NEC + mAb group (*p* = 4.01 × 10^−4^, 1.63 × 10^−4^, 5.68 × 10^−4^, 2.83 × 10^−5^, 1.58 × 10^−8^, and 4.01 × 10^−4^, respectively)**.** As observed in previous studies [31,32,33], levels of TGFβ from ileal tissue homogenates were significantly decreased in the NEC group (*p* < 0.05 vs. Ctrl + mAb). However, treatment with the anti-eNAMPT mAb significantly increased the TGFβ level (*p* < 0.01 vs. the NEC group) (Figure 6H). Together, these results are consistent with the involvement of eNAMPT/TLR4 inflammatory signaling and TGFβ in influencing genes involved in NEC pathobiology and driving disease severity.

## 4. Discussion

This work addresses the serious unmet need for therapies for NEC by targeting eNAMPT, a critical innate immunity DAMP and TLR4 ligand. Previously, the research team successfully identified and developed a humanized eNAMPT-neutralizing mAb, ALT-100, as a highly innovative therapeutic strategy to address IAI/chorio to delay premature birth and improve maternal and premature infant outcomes [49,50]. The data presented herein validate the ALT-100 mAb as a treatment modality in a rat model of NEC, an innovative approach to attenuate NEC pathophysiology.

Due to the current lack of effective NEC therapies, a number of small molecules and biologic agents targeting the inflammatory pathway and TLR4 have been suggested as potentially novel NEC therapeutic strategies [80]. Human milk [81] reduces TLR4 signaling, inhibits LPS binding to TLR4 [82], and reduces TLR4-induced NF-κB activation via epidermal growth factor activation of the PI3K–AKT pathway [39]. The TLR4 inhibitor C34, a member of a novel class of oligosaccharides, significantly reduces NEC in mice [79]. Administration of a NOD2 agonist prevents NEC through secondary inhibition of TLR4 [83]. In preclinical models, the aryl hydrocarbon receptor (AHR) ligand indole-3-carbinole (I3C), or breast milk administration, leads to the activation of AHR ligands, resulting in reduced TLR4 signaling and decreased NEC in newborn mice [84]. A systematic review and meta-analysis involving 106 infants with NEC versus controls concluded that the fecal microbiota from preterm infants with NEC has a marked increase of Proteobacteria before NEC onset [85,86], which can activate TLR4. Other studies have shown that probiotics offer a protective role by the activation of TLR9, which inhibits TLR4 signaling within the intestine via IRAK-M upregulation in mice and piglets [87,88]. Biologic therapies for NEC have been also studied, and it is suggested that anti-IL-17 and all-trans-retinoic acid supplementation restores the balance between the pro-inflammatory and the anti-inflammatory Treg cell and can reduce NEC in mice [89]. Despite these data implicating TLR4 signaling, to date, there are no NEC clinical trials that target TLR4.

This study provides compelling support for eNAMPT involvement in NEC pathobiology and eNAMPT neutralization as a novel therapeutic strategy in NEC. While the inhibition of intracellular NAMPT with FK866 has been shown to reduce NEC [90], we are the first to target eNAMPT in NEC using the eNAMPT-neutralizing mAb ALT-100. The safety of ALT-100 was validated in a completed phase 1A safety trial, and pharmacokinetic studies of a single IV-delivered ALT-100 dose (0.1–4 mg/kg) in healthy human volunteers showed the complete absence of serious adverse events, with a therapeutic half-life of 21–30 days. A phase 2A study of the IV-delivered ALT-100 mAb is currently underway for moderate-to-severe ARDS [91]. We have previously reported the effective use of the ALT-100 mAb in preclinical models of ARDS [51], radiation fibrosis [53], pulmonary hypertension [49,92], lupus vasculitis [54], NASH hepatic fibrosis and ischemia induced cardiac fibrosis [76], and chorioamnionitis [50]. Highly relevant to this study, in a preclinical pregnant mouse model of chorioamnionitis-related preterm birth (PTB), the ALT-100 mAb was shown to delay PTB, increase neonate survival, and reduce IAI-related PTB complications, including bronchopulmonary dysplasia and pulmonary hypertension [48].

Our previously reported studies demonstrate that eNAMPT neutralization rectifies inflammatory-induced gene dysregulation, consistent with reports in preclinical models of RNA sequencing, and differential gene expression analysis has identified predictable ileal tissue-specific differences between anti-NAMPT-treated and non-treated animals (Appendix A). NEC-mediated dysregulated genomic pathways are associated with innate immune responses, e.g., cytotoxicity, inflammation, NK-mediated immunity, and autoimmunity, and are consistent with the involvement of eNAMPT/TLR4 signaling-influenced genes in NEC pathobiology [93]. NEC genomic studies have confirmed that dysregulated inflammatory and apoptotic pathways are rectified by eNAMPT mAb treatment. *JunB* plays a pivotal role in the inflammatory response triggered in the NEC model, as well in cell proliferation/apoptosis [94]. Further, data showed that eNAMPT mAb treatment played a significant role in the upregulation of inflammatory cytokine repressor genes, such as *ATF3*, *FOS*, and *JUN*, and the negative regulation of TLR and the non-canonical NF-κB pathway in the NEC group (Figure 5). The current NEC genomic data are compatible with the comparative histological, molecular, and biochemical findings among treated versus non-treated NEC groups.

In addition to the decreased inflammation induced by neutralizing eNAMPT, the data presented herein suggest a novel mechanism that improves pathology in experimental NEC. First, we showed positive regulation of the TGF-β receptor signaling pathway after treatment with anti-eNAMPT in experimental NEC (Figure 6). TGFβ is a growth factor involved in numerous physiological processes, such as embryonic development, tissue repair, differentiation, and cell growth [95]. TGFβ, particularly the TGFβ2 isoform, suppresses macrophage inflammatory responses in the developing intestine and protects against inflammatory mucosal injury. TGFβ2 expression and bioactivity are decreased in NEC to lower levels than those in the premature/fetal intestine [32]. The phenotype of macrophages during NEC is strongly inflammatory and associated with increased gene expression of Smad7 and inhibition of TGFβ2 [32,96,97], which interrupts TGF-β-mediated downregulation of the pro-inflammatory response by macrophages in the NEC model [96]. Enterally administered TGFβ protects mice from experimental NEC-like injury [32]. Importantly, LPS-activated TLR4 inactivates TGFβ via crosstalk between TLR4 signaling and the canonical Wnt/β-catenin pathway [34,35,36]. In the studies presented herein, we found eNAMPT mAb alterations in both TGFβ receptor and TLR4 signaling pathways and changes in Aft3, Fos, and Jun gene expression, which are common to both pathways. Moreover, gene enrichment set analysis was performed on NEC and NEC + eNAMPT mAb-treated groups and showed that TGFβ-regulated gene sets were positively correlated with NEC + eNAMPT mAb-treated samples (Appendix A).

A limitation of this study is that it lacks data from human samples. Rodent models—while important for proof-of-concept studies—do not completely simulate human pathology. Furthermore, our model does not take into account predisposing maternal factors that can lead to NEC. For example, maternal chorioamnionitis can cause a fetal inflammatory systemic response [98], an inflammatory cascade that increases the risk for developing NEC [99]. Thus, while the results of this study are promising, further research using human tissue is necessary to validate the role of eNAMPT in NEC.

In conclusion, this research demonstrates that the DAMP eNAMPT is upregulated in an experimental model of NEC and that treatment with the eNAMPT-neutralizing mAb ALT-100 protects against NEC pathology. These findings indicate that ALT-100 has potential as a therapeutic strategy in neonates with NEC. Future studies will explore additional mechanisms involved in NEC pathology that are affected by eNAMPT.

## Figures and Tables

**Figure 1 biomedicines-12-00970-f001:**
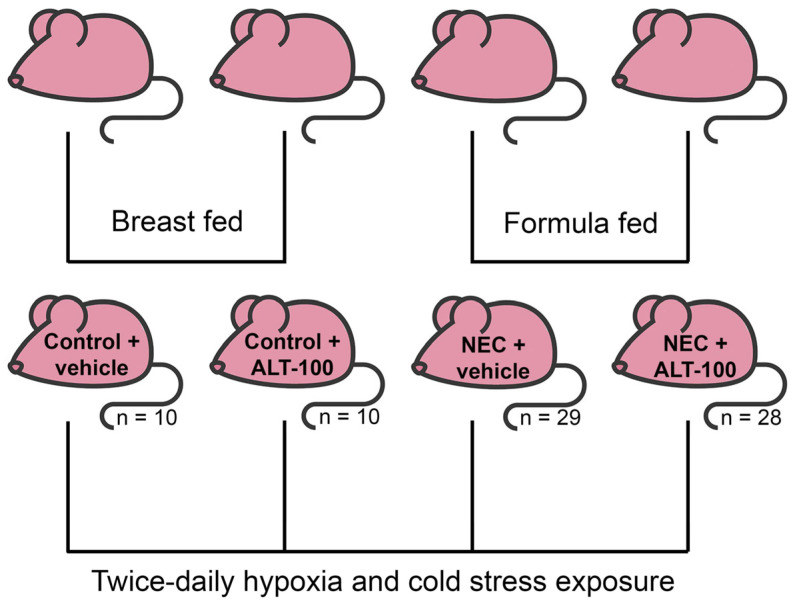
In each of the two separate studies, pups from 3–5 timed-pregnant Sprague–Dawley rats were delivered via C-section one day prior to scheduled birth. Pups were randomized and divided into four groups: control + vehicle, *n* = 10: fed by a foster dam and injected with saline (vehicle) 48 h post-delivery; control + ALT-100, *n* = 10: fed by a foster dam and injected with ALT-100 diluted in the vehicle 48 h post-delivery; NEC + vehicle, *n* = 29: exclusively hand-fed via oral gavage with formula and injected with the vehicle 48 h post-delivery; and NEC + ALT-100, *n* = 28: exclusively hand-fed via oral gavage with formula and injected with ALT-100 diluted in the vehicle 48 h post-delivery. Pups from all four groups were exposed to hypoxia (N_2_ gas for 60 s), followed by hypothermia (4 °C for 10 min) twice daily. Pups from all groups were sacrificed at 96 h post-delivery.

**Figure 2 biomedicines-12-00970-f002:**
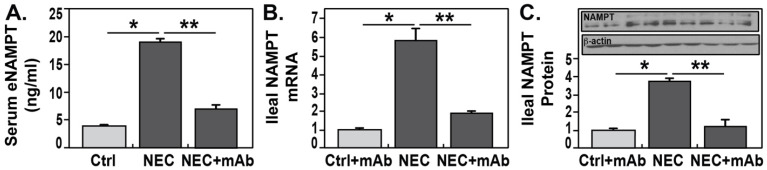
Elevated NAMPT expression in blood and ileal tissues from NEC pups. (**A**) Serum eNAMPT levels were evaluated in Ctrl + mAb (*n* = 5), NEC (*n* = 5), and NEC + mAb (*n* = 5) groups. There was a significant increase in serum eNAMPT levels among rat pups with NEC, which was attenuated in the NEC + eNAMPT ALT-100 mAb group. mAb RT-PCR (**B**) and Western blot (control 1-4 lanes; NEC 5-7 lanes; NEC + mAb 8-10 lanes) and densitometry (**C**) analyses of NAMP from ileal tissue homogenates showed a significant increase in protein NAMPT levels in the NEC group (*n* = 3) compared to the Ctrl + mAb group (*n* = 3) and a significant reduction in NAMPT expression in the NEC + ALT-100 mAb group (*n* = 3). Significant differences were determined by one-way ANOVA, followed by the Newman–Kuels post hoc test: * *p* < 0.05, ** *p* < 0.01.

**Figure 3 biomedicines-12-00970-f003:**
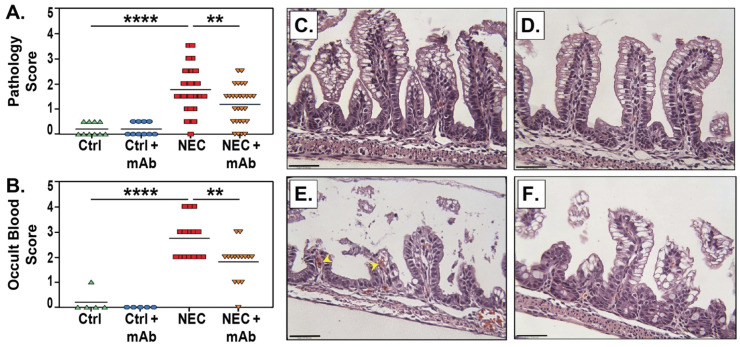
eNAMPT-neutralizing mAb reduces NEC pathology. (**A**) Pathology scores represent ileal damage, as graded on a scale of 0 (healthy) to 4 (necrosis), with half point values used for a more refined assessment of disease. A score of 2 or greater is considered NEC. Data compiled from two independent studies. Ctrl, *n* = 10; Ctrl + mAb, *n* = 10; NEC, *n* = 29; and NEC + mAb, *n* = 28. Lines represent the mean pathology score for each group. (**B**) Fecal occult blood was measured by the guaiac test in Ctrl, *n* = 5; Ctrl + mAb, *n* = 5; NEC, *n* = 15; and NEC + mAb, *n* = 15. Scores represent the level of blood in the feces, as graded on a scale of 0 to 4, and lines represent the mean occult blood score for each group. Significant differences were determined by the Kruskal–Wallis test, followed by the Benjamini–Kreiger–Yekutieli false discovery rate method for multiple comparisons (** *q* < 0.01, **** *q* < 0.0001). Representative histology images show ileal damage in Ctrl (**C**), histological damage score 0.0), Ctrl + mAb (**D**), histological damage score 0.5), NEC (**E**), histological damage score 2.5), and NEC + mAb (**F**), histological damage score 1.5). Yellow arrowheads indicate RBC accumulation in the lamina propria. Scale bars represent 50 μm.

**Figure 4 biomedicines-12-00970-f004:**
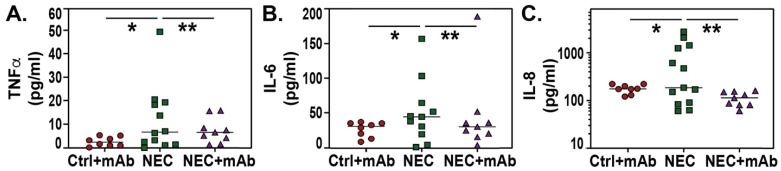
Serum proinflammatory mediators in NEC are reduced by ALT-100, the eNAMPT-neutralizing mAb. Serum samples were obtained from pups’ blood at sacrifice from Ctrl + mAb, *n* = 8; NEC, *n* = 12; and NEC + mAb, *n* = 9. TNFα (**A**), IL-6 (**B**), and IL-8 (**C**) levels were determined by ELISA. Significant differences were determined by one-way ANOVA, followed by the Newman–Kuels post hoc test (* *p* < 0.05, ** *p* < 0.01).

**Figure 5 biomedicines-12-00970-f005:**
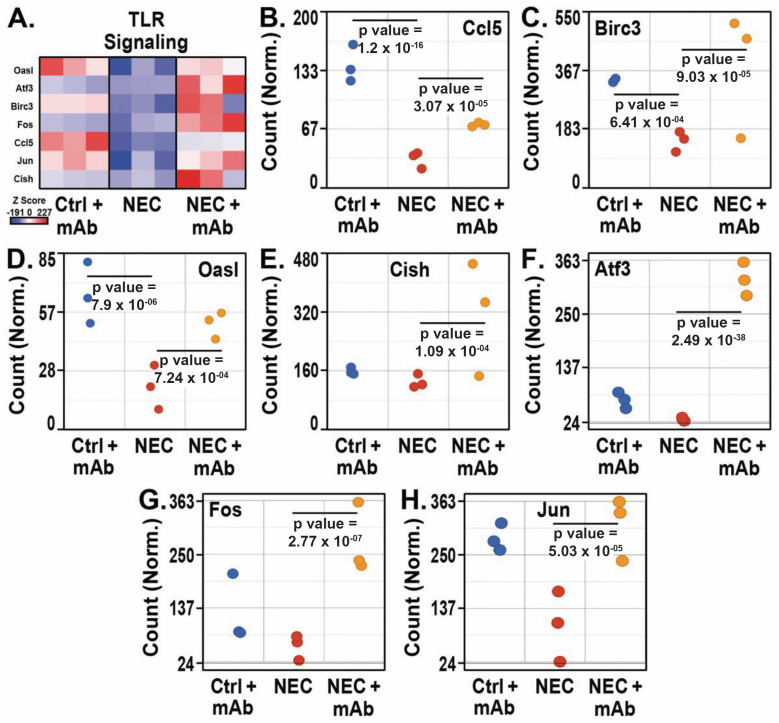
Heat map of TLR signaling genes (**A**) and TRL signaling pathway genes statistically significantly altered by the eNAMPT-neutralizing mAb (**B**–**H**). All groups, *n* = 3. Principal component analysis was performed using the Partek algorithm.

**Figure 6 biomedicines-12-00970-f006:**
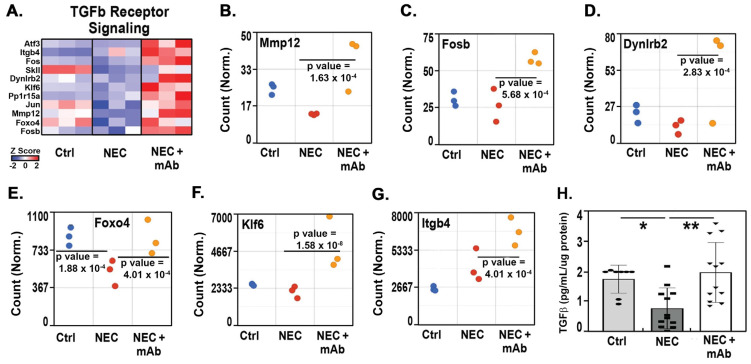
Heat map of TGF*β* signaling genes (**A**) and TGF*β* signaling pathway genes (**B**–**G**). All groups, *n* = 3. Principal component analysis was performed using the Partek algorithm. In (**H**), reduced ileal TGF*β* in the NEC groups is normalized by treatment with the anti-eNAMPT mAb (Ctrl + mAb, *n* = 8; NEC, *n* = 11; and NEC + mAb, *n* = 12). Significant differences were determined by one-way ANOVA, followed by the Newman–Kuels post hoc test (* *p* ≤ 0.05, ** *p* ≤ 0.01).

## Data Availability

Data presented are available on request from the corresponding author.

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
