# Peer review of "Extracellular Nicotinamide Phosphoribosyltransferase Is a Therapeutic Target in Experimental Necrotizing Enterocolitis"

_biomedicines, 2024, doi:10.3390/biomedicines12050970_

Round 1

Reviewer 1 Report

Comments and Suggestions for Authors

The author's team appears to have implemented similar research approaches across different diseases. This time, they turned to NEC, which seems to be a novel direction. The overall study design appears rational. Utilizing eNAMPT as a DAMP and TLR4 ligand theoretically holds promise in NEC, where TRL4 mediates inflammatory damage. However, there are several serious issues within the research:

(1) Fig 1: The baseline for Cβ-acting is uneven, and lane 4 does not show any staining. Therefore, the conclusions drawn by the authors cannot be supported. 

(2) The entire text lacks P-values.

(3) Please provide more research evidence such as the animal modeling process.

Comments on the Quality of English Language

The language is somewhat difficult to comprehend, and there are numerous instances of excessive spacing throughout the manuscript.

Author Response

14-Apr-2024

Biomedicines Editorial Office

Postfach, CH-4020 Basel, Switzerland

Re: MS ID: biomedicines-2901977

Title: Extracellular nicotinamide phosphoribosyltransferase (eNAMPT) is a novel DAMP and therapeutic target in experimental necrotizing enterocolitis

Dear Editor,

On behalf of all authors, we like to thank our reviewers for their insightful comments and have addressed and/or implemented them throughout the manuscript. Below are point-by-point responses for each reviewer’s comments in order (lines mentioned in the responses refer to where the changes occur in the clean copy).

Again, we like to thank you for re-considering our MS for the publication in Biomedicines J., special issue: Neonatal Disease; From Pathophysiology to Current and Emerging Therapeutic Approach.

Reviewer 1: “The author's team appears to have implemented similar research approaches across different diseases. This time, they turned to NEC, which seems to be a novel direction. The overall study design appears rational. Utilizing eNAMPT as a DAMP and TLR4 ligand theoretically holds promise in NEC, where TRL4 mediates inflammatory damage. However, there are several serious issues within the research:”

Comment # 1: The beta actin bands are uneven… 

Response: Thank you for noticing the error. We apologize for uploading a placeholder version of Figure 1 (now Figure 2) by mistake. The correct figure is now in place.

Comment # 2: The entire text lacks p-values

Response: Thank you for the comment. P-values have been added to the text.

Comment # 3: Animal modeling process.

Response: We apologize for the lack of clarity.  A revised description has been added to the methods section (lines 119-129) and in Results 3.1. (lines 217-221). In addition, as per Reviewer 3’s suggestion, we have added a graphical flow chart (new Figure 1) to the end of the methods section.

Comment # 4: Quality of English and excessive spacing.

Response:  Based on reviewer comments, the manuscript has been edited and spacing issues have been fixed.

Reviewer 2 Report

Comments and Suggestions for Authors

·         Title & keywords: Please do not include any abbreviations. The title needs to be clear and specific.

·     Language issues should be revised. The word “we” and “our” are frequently used throughout the manuscript (e.g., we explored, we utilized, we have shown…etc.).

·         Please define abbreviations at first use (e.g., IL-6, and TNFα).

·       Abstract: Please include the type of analysis used and significant p-value.

·       Introduction: Please expand on the risk factors and specific treatment for NEC that regulate innate immune-mediated inflammation (e.g., probiotics, breastmilk). I would recommend referring to these articles (Pediatr Res. 2018 Jan;83(1-1):16-22; Life (Basel). 2023 Feb 16;13(2):561; Adv Nutr. 2016 Sep 15;7(5):928-37).  

·         It would be also benefit to focus on the cytokine balance for preterm infants toward inflammation (e.g., IL-10, IL-6, TNF...etc.) in introduction.

·         The novelty in the last paragraph of introduction could be improved. I would suggest that the authors to present the aim of the paper with regards to what is currently known by in vivo/vitro studies, thus highlighting the added value of this study.

·         The first paragraph in the discussion is just repeated results. Please revise/delete.

·         Line 311-313: Were these all gut microbiota that may activate TLR4? Please expand on the role of gut microbiota in inhibiting/increasing intestinal inflammation.

·         Line 313-315: What about prebiotics, synbiotics, and breastmilk?

·      What are the limitations of the study? This should be in a separate section.

·    I would suggest also including a conclusion section with the consequences of the results obtained and recommendations for the future studies.

·         Please include a list of abbreviations at the end.

·         Please follow the journal guidelines for referencing.

Comments on the Quality of English Language

Moderate language editing required.

Author Response

Biomedicines Editorial Office

Postfach, CH-4020 Basel, Switzerland

Re: MS ID: biomedicines-2901977

Title: Extracellular nicotinamide phosphoribosyltransferase (eNAMPT) is a novel DAMP and therapeutic target in experimental necrotizing enterocolitis

Dear Editor,

On behalf of all authors, we like to thank our reviewers for their insightful comments and have addressed and/or implemented them throughout the manuscript. Below are point-by-point responses for each reviewer’s comments in order (lines mentioned in the responses refer to where the changes occur in the clean copy).

Again, we like to thank you for re-considering our MS for the publication in Biomedicines J., special issue: Neonatal Disease; From Pathophysiology to Current and Emerging Therapeutic Approaches.

Reviewer 2:

Comment # 1: Please do not include abbreviations in the title.

Response: Thank you for this comment. Title abbreviations have been removed.

Comment # 2: Language issues should be revised. The word “we” and “our” are frequently used throughout the manuscript.

Response: Based on reviewer comments, language throughout has been modified.

Comment # 3: Please define abbreviations at first use (e.g., IL-6, and TNFα).

Response: Thank you for this comment. IL-6 and TNFα have been properly defined when first used (lines 50-51).

Comment # 4: Abstract: Please include the type of analysis and p-value.

Response: We apologize for the lack of clarity.  P-values have been added to the abstract.

Comment # 5: Introduction: Please expand on the risk factors and specific treatments for NEC.

Response: Risk factors for NEC and the protective qualities of breastmilk have been added to the introduction (lines 64-72). Probiotics and breast milk are not treatments for NEC, but administration of them can lower the risk of NEC. As stated in the introduction, aside from surgical interventions, there are no specific treatments for NEC.

Comment # 6: It would also benefit to focus on the cytokine balance for preterm infants toward inflammation (e.g., IL-10, IL-6, TNF, etc.) in introduction.

Response: : We completely agree with this comment. Inflammatory cytokines in NEC have been added to lines 50-54.

Comment # 7: Novelty in last paragraph of introduction.

Response: Thank you for this suggestion. The sentence, These data show, for the first time, the involvement of eNAMPT in NEC pathobiology and point to eNAMPT neutralization as a potential novel strategy to address the unmet need for effective NEC therapeutics., is now the final sentence of the introduction.

Comment # 8: The first paragraph in the discussion is just repeated results. Please revise/delete.

Response: Repeating the results in the first paragraph of the discussion is a common format used in scientific writing as a means to refresh the reader’s memory. However, respecting the reviewer’s preference, that information has been deleted from the revision.

Comment # 9: Line 311-313: Were these all gut microbiota that may activate TLR4?

Response: We apologize for the mistake in lines 311-313 (353-356). Bacteroidetes and Firmicutes should not have been listed. The manuscript has been revised to state that Proteobacteria, which can activate TLR4, have been detected in increased levels in infants with NEC.

Comment # 10: Line 313-315: What about prebiotics, symbiotics, and breastmilk?

Response: Discussion of prebiotics and symbiotics, while interesting, is beyond the scope of this study. As suggested in point 5, above, information concerning breastmilk has been added to the text.

Comment # 11: Limitations of this study.

Response: Thank you for the valuable suggestion. We have added the following as the penultimate paragraph of the discussion section:

     A limitation of the current study is that it lacks data from human samples. Rodent models—while important for proof-of-concept studies— do not completely simulate human pathology. Furthermore, our model does not take into account predisposing maternal factors that can lead to NEC. For example, maternal chorioamnionitis can cause fetal inflammatory systemic response [94], an inflammatory cascade that increases the risk for developing NEC [95]. Thus, while the results of the current study are promising, further research using human tissue is necessary to validate the role of eNAMPT in NEC.

Comment # 12: Please add conclusion and recommendation for future studies.

Response: The last paragraph of the discussion has been modified to a conclusion and future studies.

Comment # 13: Please follow the journal guidelines for referencing.

Response: The journal’s guide states that any format for references can be used as long as it remains constant throughout. This is the case for the reference style used in this manuscript.

Reviewer 3 Report

Comments and Suggestions for Authors

Thank you very much for allowing me to review the work entitled "Extracellular nicotinamide phosphoribosyltransferase" (biomedicines-2901977), submitted to the Section "Cell Biology and Pathology" in the Special Issue "Neonatal Disease: From Pathophysiology to Current and Emerging Therapeutic Approaches".

This study addresses a significant issue, namely necrotizing enterocolitis (NEC), which stands as the most common gastrointestinal emergency among premature infants. It explores the potential involvement of eNAMPT (extracellular nicotinamide phosphoribosyl transferase), a novel DAMP (damage-associated molecular pattern) and TLR4 ligand, in NEC pathobiology, utilizing an eNAMPT-neutralizing monoclonal antibody (ALT-100).

Comments:

The abstract should indicate the design and methodology employed, as well as the main results quantitatively. It is important for the abstract to be as comprehensive as possible so that potential readers have the most complete information about the work undertaken.

The introduction effectively sets up the hypotheses and objectives of the study. The bibliography is adequate. However, it would be insightful to know the frequency of this condition in the pediatric population. The hypothesis should be more specific, as well as the objectives; the term "explore" is overly vague.

In the methodology section, the experimental design should be stated first. Additionally, it is noted that the reagents are sourced from a pharmaceutical company (Sigma-Aldrich, St. Louis, MO...). Typically, such information is not explicitly provided; instead, the type of reagent used is mentioned. It may be worth consulting with the editorial team regarding the appropriateness of featuring pharmaceutical brand names in the article. I suggest that a flowchart be included for the described protocol to enhance understanding of the procedure and the number of animals involved in the experiment.

Regarding the statistical analysis section, non-parametric tests are described, followed by a mention of comparison using ANOVA, which is suitable for normally distributed data. Could you please clarify this aspect?

Results sections typically present the findings of the experiments leading to the study. In this article, however, references to the literature, more suitable for the discussion section, are included in the results section (lines 181-184).

On line 263 of the results section, the analysis of principal components is explained, although this analysis is not described in the statistical methodology.

The discussion is well-presented and highly engaging. I would only suggest evaluating the strengths and weaknesses of the experiment presented, not only focusing on it being a mouse model but also considering other relevant factors.

Author Response

Biomedicines Editorial Office

Postfach, CH-4020 Basel, Switzerland

Re: MS ID: biomedicines-2901977

Title: Extracellular nicotinamide phosphoribosyltransferase (eNAMPT) is a novel DAMP and therapeutic target in experimental necrotizing enterocolitis

Dear Editor,

On behalf of all authors, we like to thank our reviewers for their insightful comments and have addressed and/or implemented them throughout the manuscript. Below are point-by-point responses for each reviewer’s comments in order (lines mentioned in the responses refer to where the changes occur in the clean copy).

Again, we like to thank you for re-considering our MS for the publication in Biomedicines J., special issue: Neonatal Disease; From Pathophysiology to Current and Emerging Therapeutic Approaches.

Reviewer 3: “The study addresses a significant issue, namely necrotizing enterocolitis (NEC), which stands as the most common gastrointestinal emergency among premature infants.”

Comment # 1: The abstract should indicate the design and methodology employed, as well as the main results quantitatively. It is important for the abstract to be as comprehensive as possible so that potential readers have the most complete information about the work undertaken.

Response: Thank you for this valuable recommendation. The abstract has been modified as suggested.

Comment # 2: The introduction effectively sets up the hypotheses and objectives of the study. The bibliography is adequate. However, it would be insightful to know the frequency of this condition in the pediatric population. The hypothesis should be more specific, as well as the objectives; the term "explore" is overly vague,

Response: Thank you for this suggestion. As suggested, the frequency of NEC has been added to the first paragraph of the introduction (lines 36-37) We have also added a sentence to the last paragraph of the introduction that states our hypothesis.

Comment # 3a: In the methodology section, the experimental design should be stated first. Additionally, it is noted that the reagents are sourced from a pharmaceutical company (Sigma-Aldrich, St. Louis, MO...). Typically, such information is not explicitly provided; instead, the type of reagent used is mentioned. It may be worth consulting with the editorial team regarding the appropriateness of featuring pharmaceutical brand names in the article.

Response: Thank you for this suggestion. Reagents are offered as part of the experimental design and is offered first in the methods section because it includes vendor and catalog information on antibodies used throughout the studies. This is important to include not only so it is clear what we used and where we purchased these antibodies, but also so other researchers could replicate our work. For clarity, we have changed the name of the sub-section from “Reagents” to “Antibodies.”

Comment # 3b: I suggest that a flowchart be included for the described protocol to enhance understanding of the procedure and the number of animals involved in the experiment.

Response: Thank you for this great suggestion. A flow chart depicting the experimental animal protocol has been added (new Figure 1) with the number of animals per group shown.

Comment # 4: Regarding the statistical analysis section, non-parametric tests are described, followed by a mention of comparison using ANOVA, which is suitable for normally distributed data. Could you please clarify this aspect?

Response: Only NEC pathology and occult blood in feces were analyzed using a nonparametric test (Kruskal-Wallis). The sentence in the original submission, Continuous data were compared using nonparametric methods and categorical data by chi square test… was incorrect. The sentence now reads, All other data were analyzed using standard one-way ANOVA and groups were compared using the Newman-Keuls test.

Comment # 5: Results sections typically present the findings of the experiments leading to the study. In this article, however, references to the literature, more suitable for the discussion section, are included in the results section (lines 181-184).

Response: We respectfully disagree with the comment regarding references are not appropriate for the results section. It is common convention to reference relevant literature in results when they bring clarity to why specific experimental groups were used. The references we believe you referred to are first presented in the methods section and we chose to reintroduce the specifics of the animal model in the result section 3.1 because of the third comment of Reviewer 1, which asked for clarity regarding the animal model process.

Comment # 6: On line 263 of the results section, the analysis of principal components is explained, although this analysis is not described in the statistical methodology.

Response: We apologize for not including the information. This sentence was added to the methodology statistical section: “PCA was derived based on maximum variance in transcriptomic data and generated by the Partek Algorithm”.

Comment # 7: Strengths and weaknesses of the studies:

Response: As also suggested by the previous reviewer (see Reviewer 2, comment #11), we have added a new paragraph with study limitations to the discussion.

Round 2

Reviewer 2 Report

Comments and Suggestions for Authors

No further comments.

Comments on the Quality of English Language

N/A

Author Response

Based on reviewer comments, the manuscript has been edited and spacing issues have been fixed.

Reviewer 3 Report

Comments and Suggestions for Authors

I have thoroughly reviewed the new version of the manuscript "Extracellular nicotinamide phosphoribosyltransferase" (biomedicines-2901977), as well as the authors' response to the comments and proposed improvements to the article.

Regarding the indication that the abstract should present the design and methodology employed, the authors state that they have done so, but these aspects are not evident.

The objective stated in the abstract remains "to explore," which is overly vague.

The authors claim to have included a clearer statement of the objective in lines 36-37 of the introduction, but these lines correspond to the keywords.

The methodology section should specify the applied design.

Traditionally, the hypothesis and objective are presented at the end of the introduction; however, the authors do not provide this information. Line 114-116 of the introduction presents results, which is not customary in introductions.

Author Response

Comment #1: Regarding the indication that the abstract should present the design and methodology employed, the authors state that they have done so, but these aspects are not evident.

Response: More explicit language concerning the hypothesis has been added to the abstract (lines 13 – 17 in the new markup copy). Regarding the methodology presented, the abstract in the first resubmission contained a detailed description of the animal model and the experimental groups, as well as a mention of the other, more standard techniques used. We believe the methods are abstract-appropriate, especially given the 200-word abstract limit found in the Biomedicines’ Instructions to Authors.

Comment #2: The objective stated in the abstract remains "to explore," which is overly vague.

Response: The sentence using the word “explore” has been deleted (line 17-19 in the new markup copy) and rewritten.

Comment #3: The authors claim to have included a clearer statement of the objective in lines 36-37 of the introduction, but these lines correspond to the keywords.

Response: We apologize for the confusion. In the introductory statement in the previous responses to the reviewers, we indicated that “lines mentioned in the responses refer to where the changes occur in the clean copy”. The line numbers cited by the reviewer corresponded to the original markup version. We have now more clearly stated the objective in lines 91- 95 in the new markup copy.

Comment #4: The methodology section should specify the applied design.

Response:  We queried the editor to ask for clarification from the reviewer with regard to what was meant by the comment, and the section managing editor responded with the following:

Does the phrase "The methodology section should specify the applied design" refer to a paper or report that discusses the methods used in the research must clearly describe the design of the study that was implemented? Here's a breakdown of what this entails:
     Methodology Section: This is a part of a research paper or proposal that explains the methods used to conduct the research.
     Specify: To specify means to describe in detail or to make clear. It implies that the description should be precise and leave no room for ambiguity.
     Applied Design: This refers to the plan or blueprint that was used to structure the research. It includes the type of study (e.g., experimental, observational, survey, etc.), the subjects or data sources, the procedures followed, and the rationale behind these choices.

As the methods section states the type of study (experimental), data sources (premature rats, specific tissues/products from said rats), procedures followed (protocol for inducing NEC, histology, ileal pathology scoring, ELISA, RNAseq, etc.. We have, however, added the following regarding the applied design:

“The formula/hypoxia/hypothermia model of NEC was chosen because it most closely mimics the risk factors for NEC: prematurity, formula feeding, and stress (modeling apnea of prematurity, a common condition of premature infants, and the inability of preemies to adequately temperature regulate)”.

This addition—with appropriate references—can be found in lines 120 – 124 in the new markup copy.

Comment #5: Traditionally, the hypothesis and objective are presented at the end of the introduction; however, the authors do not provide this information. Line 114-116 of the introduction presents results, which is not customary in introductions.

Response: The hypothesis was stated in the first sentence of the last paragraph of the introduction in the resubmission: “Because eNAMPT has been shown to activate TLR4—which is crucial to NEC development—we hypothesize that treatment with an eNAMPT-neutralizing mAb would protect against NEC.” We agree, however, that a clear objective was not provided. We have added this information to lines 91- 95 of the new markup copy.

We respectfully disagree that results in the introduction are not customary. In our experience, they are commonly summarized at the end of the introduction. However, because the three currently published articles for this special issue do not utilize this stylistic format, we have removed the summation of findings (lines 96 -100 of the new markup copy).

Round 3

Reviewer 3 Report

Comments and Suggestions for Authors

Re: MS ID: biomedicines-2901977

Title: Extracellular nicotinamide phosphoribosyltransferase (eNAMPT) is a novel DAMP and therapeutic target in experimental necrotizing enterocolitis.

I have thoroughly reviewed the revised version (v.3) of the manuscript, as well as the authors' response to the requested suggestions and clarifications. I find that most of the suggestions have been incorporated, resulting in a clearer and more concise work. However, further research will be necessary to verify the involvement of eNAMPT in NEC pathobiology and the potential of eNAMPT neutralization as a therapeutic strategy for addressing the unmet needs in NEC treatment.